# Influence of Duty Ratio and Current Mode on Robot 316L Stainless Steel Arc Additive Manufacturing

**Ping Yao** [1,2]**, Hongyan Lin** [1,*]**, Wei Wu** [3,*] **and Heqing Tang** [1]

1   School of Mechatronic Engineering, Guangdong Polytechnic Normal University, Guangzhou 510635, China; gsyaop@gpnu.edu.cn (P.Y.); happythq2015@163.com (H.T.)
2   Department of Mechatronic Engineering, Guangzhou Institute of Science and Technology, Guangzhou 510540, China
3   School of Automobile and Transportation Engineering, Guangdong Polytechnic Normal University, Guangzhou 510635, China
*   Correspondence: sjy843644862@163.com (H.L.); wuwei_5v@gpnu.edu.cn (W.W.); Tel.: +86-18316563881 (H.L.); +86-15018759451 (W.W.)

**Abstract:** Wire and arc additive manufacturing (WAAM) is usually for fabricating components due to its low equipment cost, high material utilization rate and cladding efficiency. However, its applications are limited by the large heat input decided by process parameters. Here, four 50-layer stainless steel parts with double-pulse and single-pulse metal inert gas (MIG) welding modes were deposited, and the effect of different duty ratios and current modes on morphology, microstructure, and performance was analyzed. The results demonstrate that the low frequency of the double-pulse had the effect of stirring the molten pool; therefore, the double-pulse mode parts presented a bigger width and smaller height, finer microstructure and better properties than the single-pulse mode. Furthermore, increasing the duty ratio from 35% to 65% enlarged the heat input, which then decreased the specimen height, increased the width, and decreased the hardness and the tensile strength.

**Keywords:** welding mode; single-pulse; double-pulse; duty ratio; mechanical property

## 1. Introduction

Additive manufacturing (AM) has received more attention and application in the manufacturing industry recently. However, wire and arc additive manufacturing (WAAM) is more suitable for manufacturing complex components than an electron beam and laser [1]. At the same time, it has the advantages of low equipment and operational costs, and high material utilization rate and cladding efficiency [2]. Therefore, WAAM based on gas metal arc welding (GMAW) has become the research focus of many scholars and institutions in China and abroad. Zhang et al. [3] carried out a preliminary study of GMAW rapid formation, while Jiang et al. [4] developed a cold metal transfer (CMT) deposition method for manufacturing aluminum alloy. The results of the above studies showed that back-and-forth depositing could reduce the heat accumulation and difference of two endings, and thereby improving the forming efficiency and reducing roughness. Furthermore, Montevecchi et al. [5] employed a GMAW method to manufacture H08Mn2Si deposited samples and research the influence of cooling time on the morphology. The above studies indicated that WAAM based on GMAW could manufacture the metal parts and different parameters had an effect on the morphology.

In recent years, countless scholars have done researches on stainless steel arc additive manufacturing. Wang et al. [6] studied the quality of 316L stainless steel by two modes of high-speed pulse and high-speed arc. The results showed that the high-speed arc had higher tensile strength and hardness than the high-speed pulse because of its lower heat input and finer crystal structure, while Chen et al. [7] made a 316L austenitic stainless steel part with high power metal inert gas (MIG) arc AM and found that with the increase of

arc power, the tensile strength, yield strength and area reduction of 316 samples decreased. In addition, Xiao et al. [8] studied the effect of thermal behavior on its microstructure evolution in tungsten inert gas (TIG) arc additive manufacturing of 316L stainless steel. In the process of microstructure evolution in the non-equilibrium state of heat accumulation, the increasing heat accumulation made the undercooling change during solidification, and the microstructure of the lower deposition layer gradually changed from skeleton ferrite to dendrite. Furthermore, Wang et al. [9] prepared 316L austenitic stainless steel by CMT. It was found that the remelting zone had higher ferrite content and smaller austenite dendrite size, and more dispersed orientation and lower residual stress than the overlapping zone. Similarly, Su et al. [10] used the solidification mode of austenitic stainless steel to predict the microstructure of additive manufacturing. The results showed that the predicted microstructure was consistent with the actual microstructure, which was dendritic austenite, lath ferrite and skeletal ferrite, and the microhardness of the lower part was higher than that of the upper part. While Wang et al. [11] have formed 316L austenitic stainless steel thin-walled parts by the CMT arc additive manufacturing method. The microstructure of the parts was $\gamma$-Fe and $\delta$-ferrite, and the morphology of $\delta$-ferrite was dendritic and vermicular. The microhardness results showed that there was little change in the hardness perpendicular to and parallel to the deposition direction, which was related to the uniformity of the microstructure formed in all directions.

Although using the same average current, different welding current processes cause different current waveforms, voltages, and wire feeding speed; that is, the thermal input is altered, which can further influence the forming and performance of the deposition. In reference [12], ER70S-6 mild steel was built by means of direct current MIG, cold metal transfer (CMT), CMT advanced polarity and CMT continuous trajectory welding. The morphology, microstructure, and hardness of deposition were analyzed, and then the tensile properties were estimated. The results revealed that CMT continuous trajectory deposition was the best in forming and properties. Rodriguez et al. [13] performed depositions of 316L using continuous and pulse current by CMT and top tungsten inert gas welding and compared the forming efficiency and surface roughness. The results showed that pulse current was better than other welding, which had higher molding efficiency and required tensile properties.

Through the previous analysis study of weld morphology, Yao et al. [14–16] got a visual understanding of the welding quality, then Wu et al. [17,18] found that the morphology of the part could be improved by asynchronous arc starting and extinguishing. Moreover, Liu et al. [19] studied the influence of arc AM process parameters on the forming quality through an orthogonal experiment. The results showed that the cooling between passes was sufficient and the overlapping parts were obviously divided, which lead the hardness on both sides of the weld bead to be high, whereas on the incomplete recrystallization area, it was low. Lin et al. [20] studied the influence of welding speed and interlayer cooling time on the morphology of the AM, and found that the welding speed had a more obvious influence on the morphology. When the interlayer cooling time reached a certain value, the interlayer temperature was constant, and the morphology changed little.

The duty ratio in double-pulse waveform means the percentage of a high level in a pulse period. This means that the peak time is different as well as the duty cycle in a constant period. In other words, the average current in a period is different, which finally affects the heat input. However, there are few researches on 316L stainless steel additive manufacturing with different duty ratios of the MIG welding double-pulse. Therefore, in this paper, a comparative study of double-pulse enhanced welding with different pulse duty ratios and a single-pulse of stainless steel additive manufacturing was proposed by a robot-manufacturing platform. Then the influence of a double-pulse and duty ratio process on the additive manufacturing molding, microstructure, microhardness and tensile properties was analyzed. This study will provide an alternative additive manufacturing process for grain refinement and performance improvement and lay a foundation for further research and application of double-pulse welding arc additive manufacturing.

## 2. Materials and Methods

*Test Materials and Equipment*

As shown in Figure 1, the MIG welding arc additive manufacturing system is composed of an S5-RoboMIG welding machine (LORCH, Auenwald, Germany), RF-06 wire feeder (LORCH, Auenwald, Germany), FANUC M-10iA robot (FANUC, Tokyo, Japan), multi-source information acquisition (self developed) and industrial camera (Mecaweld Technology LLC, Zhuhai, China). A 304 steel plate of $250 \times 100 \times 5$ mm$^3$ was used as the test substrate, and 316L austenitic stainless steel wire with 1.2 mm diameter was selected for this test. The wire and plate composition is presented in Table 1. Before the welding, the oxide film on the steel plate surface should be polished off with sandpaper.

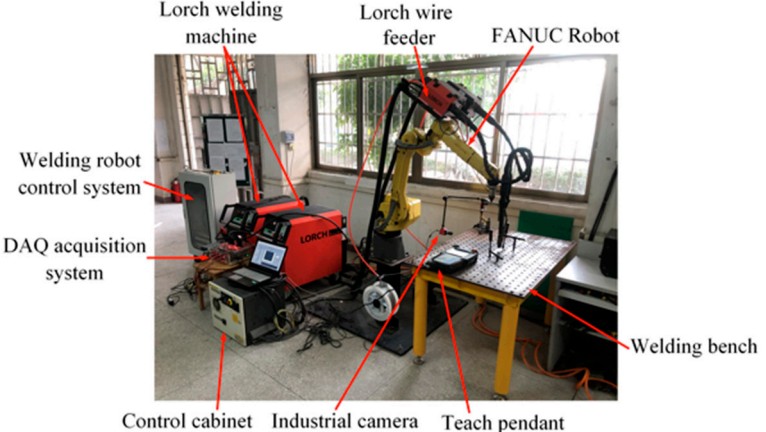

**Figure 1.** Robot arc additive manufacturing platform.

**Table 1.** Chemical composition of the welding wire and substrate (mass fraction, %).

| Material | C | Si | Ni | Cr | Mn | P | Mo | S |
|---|---|---|---|---|---|---|---|---|
| 316L | ≤0.03 | ≤1 | 10~14 | 18~20 | ≤2 | ≤0.03 | 2~3 | ≤0.03 |

The test process is shown in Figure 2. Four groups of different process parameters were used to manufacture four 50-layer walls with 80 mm length and reciprocating depositing method proposed by Xiong [21]. During the AM process, the welding torch was vertical to the plate and the distance between the nozzle and the substrate was 15 mm. The process parameters are shown in Table 2, and the welding speed was 30 cm/min and the layer cooling time was 40 s. During the deposition, 99% pure argon with a flow rate of 20 L/min was used. As the substrate would produce the residual stress and be easy to warp after deposition, it is necessary to fix the substrate with a fixture, which could prevent the deformation effect on the formation from the substrate.

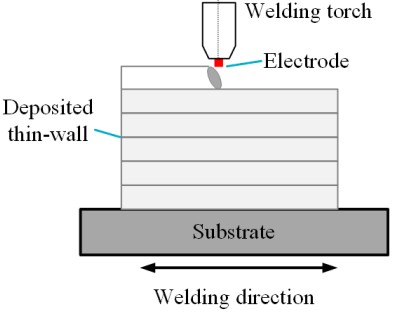

**Figure 2.** Schematic diagram of the robot additive manufacturing process.

**Table 2.** Forming process parameters of four depositions.

| Number | Current Mode | Average Current (A) | Duty Ratio |
|--------|--------------|---------------------|------------|
| A | Double-pulse strengthen | 80 | 35% |
| B | Double-pulse strengthen | 80 | 50% |
| C | Double-pulse strengthen | 80 | 65% |
| D | Single-pulse | 80 | - |

After deposition, the metallographic and tensile specimens were extracted from the middle part and cut by a wire cutting machine, as shown in Figure 3a. Furthermore, the tensile samples designed in the test were smaller than the minimum size of the international standard sample. Therefore, all tensile sample sizes were reduced in proportion to the international standard; the tensile sample size is shown in Figure 3b. Then the metallographic samples were polished with 800#, 2000#, 3000#, and 5000# sandpapers. After that, the specimens were polished with w2.5 and w0.5 diamond polishing agents until the mirror surface was smooth. Finally, the specimens were etched with corrosion solution aqua regia (HCl:HNO$_3$ = 3:1) for 30 s, and the microstructure was observed by Leica DM 2700 M (Leica Microsystems CMS Gmbh, Wetzlar, Germany). Then HR-150DT electric Rockwell hardness tester (Shanghai SHANGCAI Tester machine Co., Ltd., Shanghai, China) was used for hardness measurement with a load of 0.5 kg and maintained for 5 s. The hardness was measured every 1.5 mm from the bottom to the top along the deposited direction on both sides and the middle, as shown in Figure 3c. The WEW-6003 microcomputer hydraulic universal testing machine (Shanghai HUALONG Testing Instrument Co., Ltd., Shanghai, China) was used as the tensile equipment with a speed of 0.02 kN/s, and the average value of three tensile specimens was calculated to ensure the accuracy.

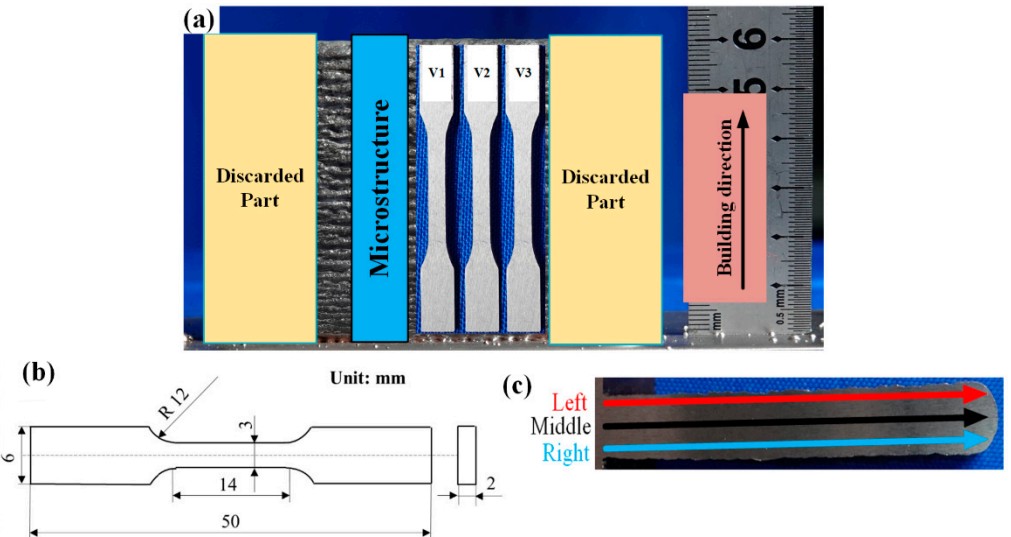

**Figure 3.** Sampling size and measuring position: (**a**) Extraction locations of microstructure and tensile samples on each wall; (**b**); Tensile specimen sizes; (**c**) Test position of hardness.

## 3. Results and Discussion

### 3.1. Morphology Analysis

During the manufacturing process, the arc currents were all larger than 40 A and repeated well in each cycle, and the arc would not be extinguished, which indicated that the double-pulse strengthening and single-pulse additive manufacturing process had a good droplet transfer mode. Therefore, double-pulse strengthening and single-pulse current modes are all stable additive manufacturing processes.

After forming and waiting for the wall to cool to room temperature, six points on the wall were selected for testing the height and width by a vernier caliper with an accuracy

of 0.02 mm, as shown in Figure 4. Each point was measured three times to calculate the average value of the wall. Figure 5 shows the appearance of the four forming parts. It can be observed that the surface of the samples was well formed and without obvious welding defects, and the layer bonding was relatively smooth. However, a greater amount of spatter can be seen on the substrate in Figure 5d, which indicates that the stability of the single-pulse deposition process is worse than that of the double-pulse deposition process, and the forming roughness is bigger.

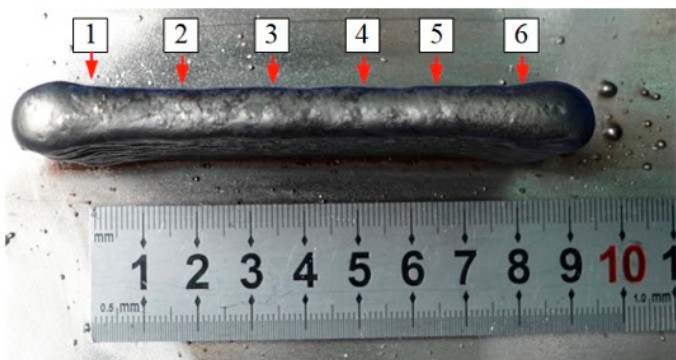

**Figure 4.** The schematic diagram of measuring position in the parts: numbers 1 to 6 represent the measuring position of the width and height.

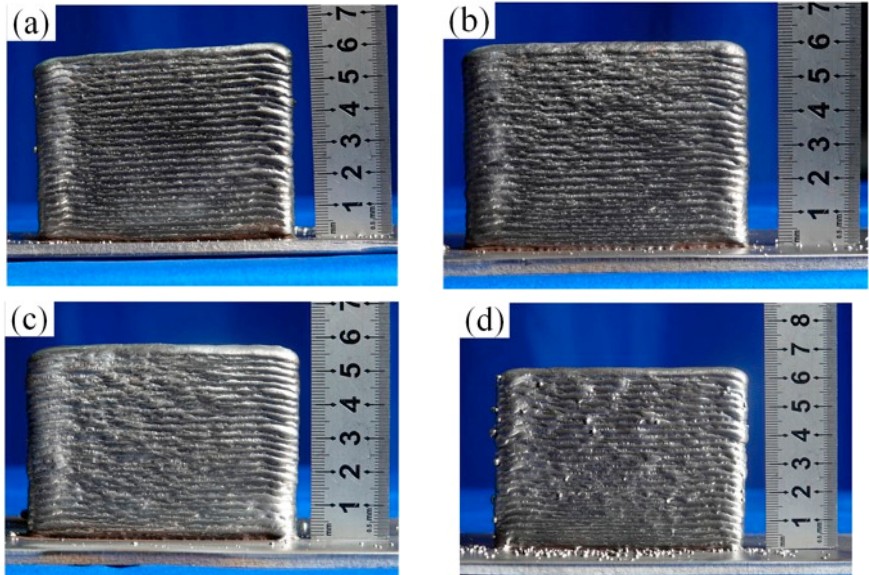

**Figure 5.** The morphology of four additive manufacturing walls: (**a**) deposited part A, (**b**) deposited part B, (**c**) deposited part C, (**d**) deposited part D.

It can be seen from Figures 6 and 7 that the average height of A, B, C and D part was $61.92 \pm 0.93$, $61.07 \pm 0.48$, $58.62 \pm 0.20$ and $63.52 \pm 0.76$ mm, respectively, and the average width was $7.63 \pm 1.41$, $8.73 \pm 1.38$, $9.95 \pm 1.08$ and $7.67 \pm 0.67$ mm, respectively; however, the height of single-pulse part D was bigger than that of double-pulse ones. Furthermore, with the increase of the duty ratio, the height decreased, while the width of double-pulse parts increased obviously. This was because the increasing duty ratio could increase the welding current, which enlarged the cladding amount of the molten pool per second with the constant welding speed. Furthermore, the stirring effect of the double-pulse could increase the molten pool width and decrease the layer height.

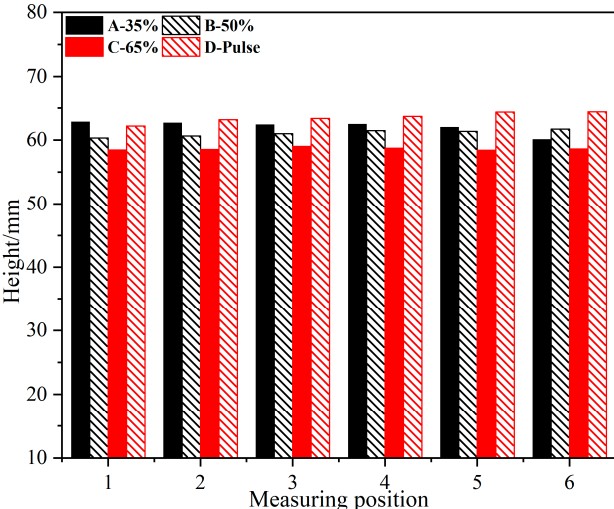

**Figure 6.** Six-point heights of four parts.

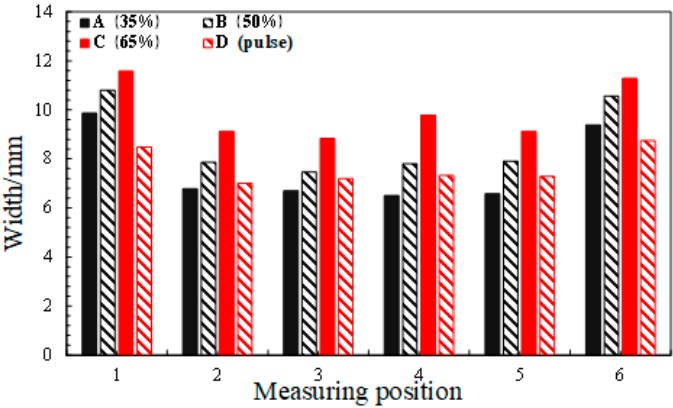

**Figure 7.** Six-point widths of four parts.

### 3.2. Microstructure

The macrostructure of 50 deposited layers in the middle of specimen B is shown in Figure 8. As a result of the stable deposition process, the intervals between layers are clearly visible, and the layer height was uniform and was about 1.2 mm. Furthermore, the interlayer metal was well connected, and no macroscopic defects such as visible pores, cracks, and unfused layers are visible in the depositions.

Figure 9 shows that the microstructure of different layers was mainly composed of austenite and ferrite [22,23]. The number of secondary dendrite arms and distance could be counted to calculate the secondary dendrite spacing, which represented the average size of the grains [17]. The secondary dendrites distance of A, B, C and D were 16.73 ± 1.88, 19.84 ± 1.57, 20.71 ± 0.65 and 22.91 ± 1.04 μm, respectively. As the heat input was increased with the duty ratio changing from 35% to 60%, the microstructure became coarser, however, because of the stirring effect of double-pulse on the molten pool [24], which meant the microstructure of the double-pulse specimens was finer than the single-pulse one. During the deposition process, the temperature along the deposition direction changed fast, therefore, the grain was mainly columnar and its growth direction was perpendicular to the fusion line of each layer [25].

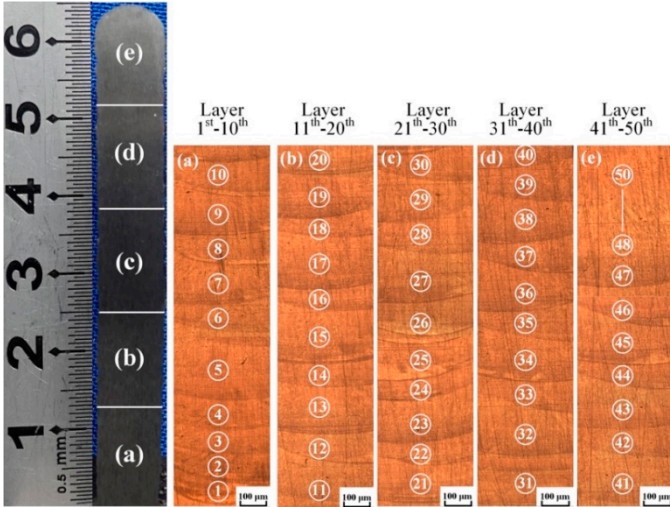

**Figure 8.** Macroscopic metallography of different layers in the middle section of specimen B: (**a**) layers 1–10, (**b**) layers 11–20, (**c**) layers 21–30, (**d**) layers 31–40, (**e**) layers 41–50.

It can be seen from the figures that the microstructure in the middle part of the layer was arranged in a skeleton shape, and the growth direction was obviously upward. This was because during the solidification process, the temperature gradient was larger and the crystallization rate was smaller at the melting boundary of the deposition. With the transition from the melting boundary to the center of the deposition layer, the temperature gradient gradually decreased and the crystallization rate increased, and the crystal morphology would develop to the columnar and equiaxed crystals. Finally, a large number of skeleton ferrite would be formed in the middle of the deposited layer.

**(a)**

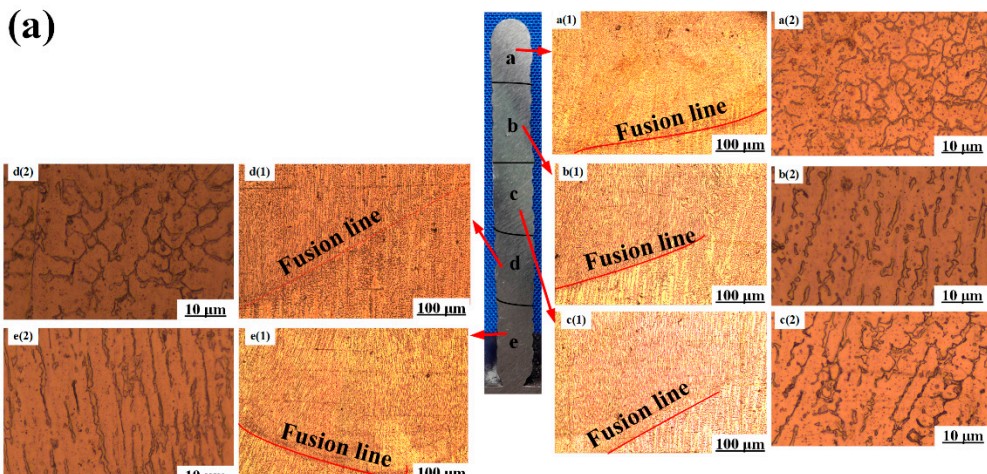

**Figure 9.** *Cont.*

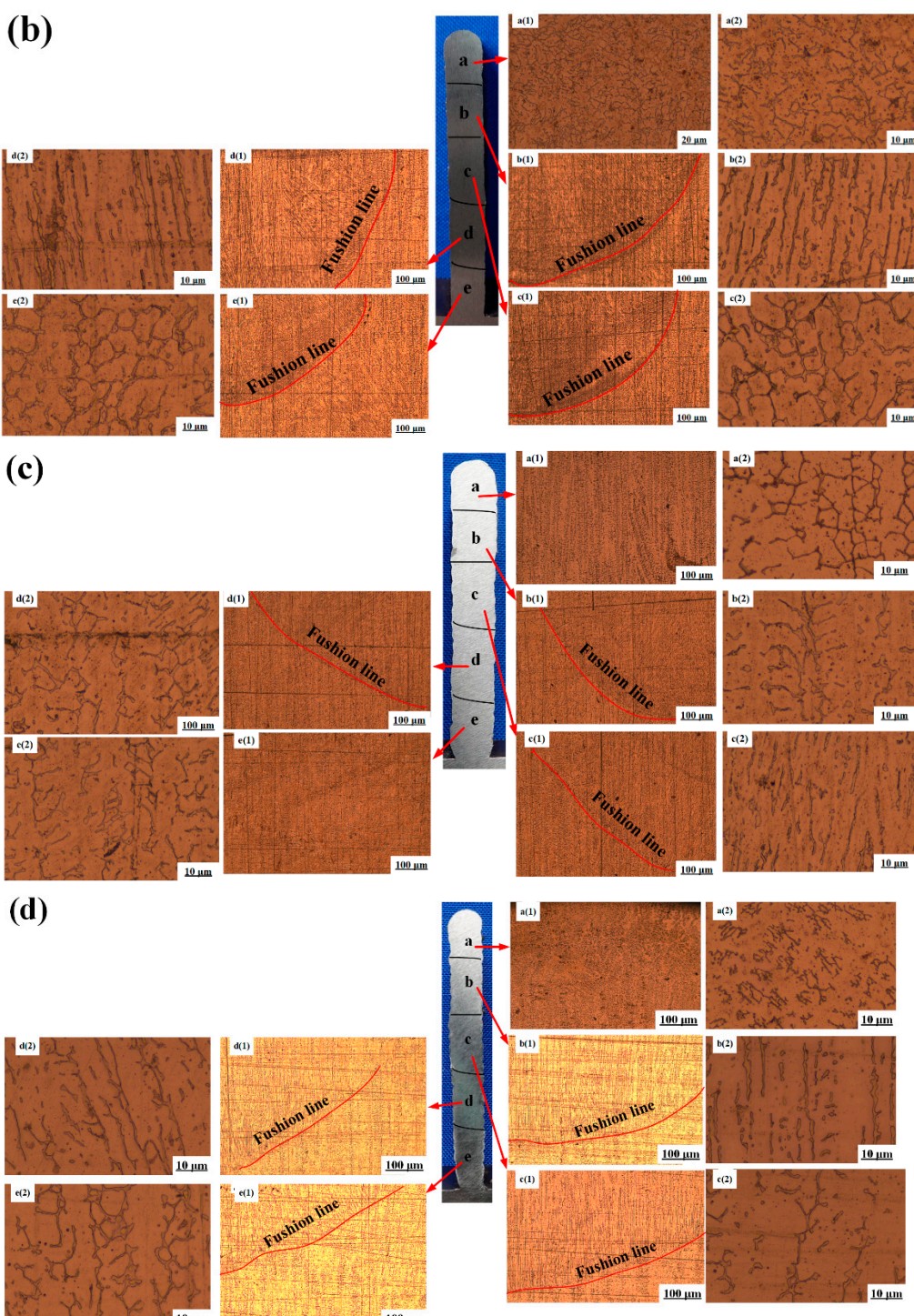

**Figure 9.** Microstructure of different layers of specimens: (**a**) specimen A, (**b**) specimen B, (**c**) specimen C, (**d**) specimen D. (**a1,a2**) layers 41–50, (**b1,b2**) layers 31–40, (**c1,c2**) layers 21–30, (**d1,d2**) layers 11–20, (**e1,e2**) layers 1–10.

*3.3. Mechanical Properties*

Figure 10 shows the Rockwell hardness values for four parts. The average Rockwell hardness of four specimens was 51.84 ± 1.79, 51.2 ± 1.71, 51.18 ± 1.18, and 50.3 ± 1.87 HRA, respectively. With the increase of the duty ratio, the heat input increased and the hardness decreased, which corresponded to the grain size and followed the Hall–Petch formula; that is, the finer the grain size, the larger the hardness. However, because of the stirring effect of double-pulse on the molten pool, the microstructure was refined [24], and

the double-pulse hardness was higher than the single-pulse one. Furthermore, the hardness of the bottom layers was larger than the top layers, and the hardness of both sides was smaller than the middle parts, this was for the higher cooling rate of the bottom part from the base plate and the sides from the atmosphere, then the grain size of the bottom and sides was finer than others, which was the same result as in Reference [18]. In addition, the bottom layer sustained the transient thermal cycles from the depositions of the following layers; therefore, the bottom part had higher hardness.

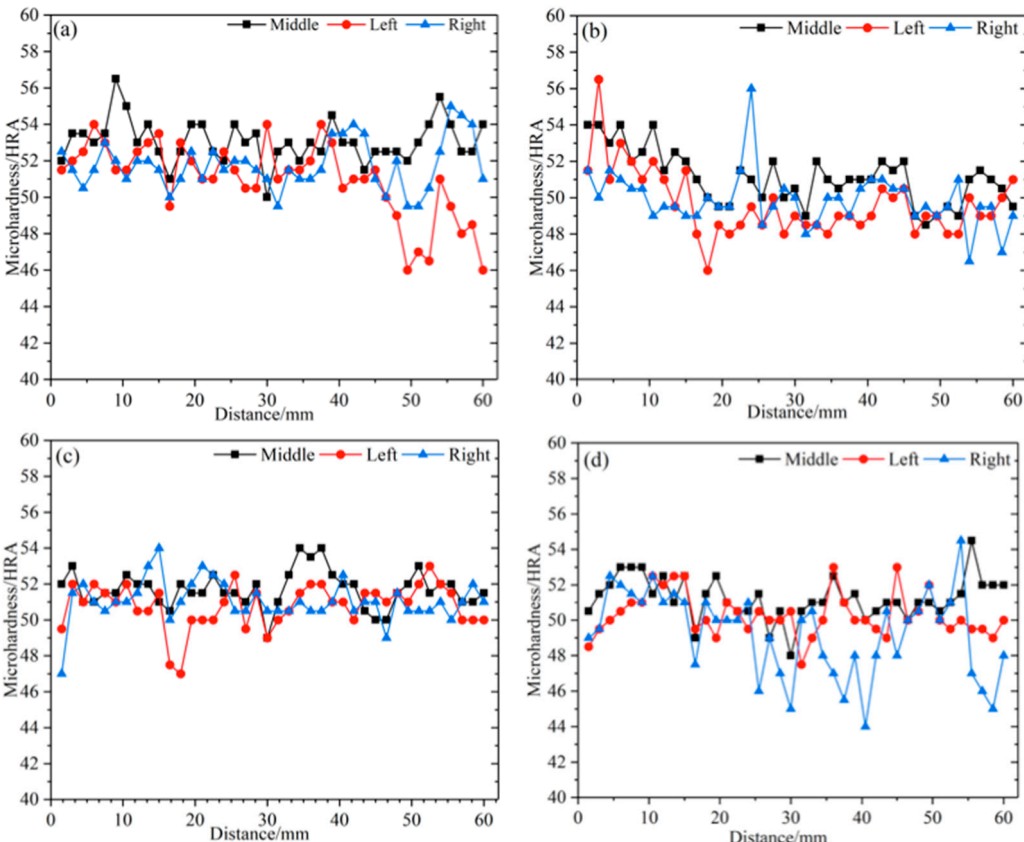

**Figure 10.** The hardness curves of four specimens: (**a**) deposited part A, (**b**) deposited part B, (**c**) deposited part C, (**d**) deposited part D.

The stress-displacement curves of four parts are displayed in Figure 11. The failure strain of parts A, B, C and D was 7.56, 10.92, 8.47 and 9.13 mm, respectively, and the four average values of ultimate tensile strength were 705 ± 5, 687.5 ± 2.5, 672.5 ± 2.5, 632.5 ± 12.5 MPa, respectively. The results showed that with the increase of the duty ratio, the tensile strength decreased, and the tensile strength and the elongation of the specimens with double-pulse were obviously higher than the one with single-pulse, which corresponded to the Rockwell hardness and the grain size. That is, the lower heat input increased the crystal-cooling rate, which improved the tensile strength and hardness. These values are higher than the industrial forging standards of 450 MPa [26], and slightly lower than the average values for the substrate of 641.2 MPa, except for the single-pulse one, and the overall properties were more satisfactory.

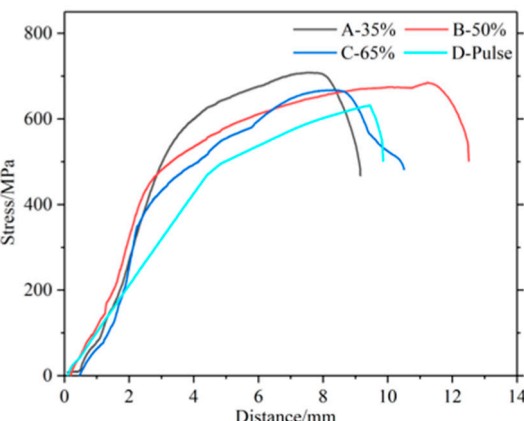

**Figure 11.** Stress–displacement curves of four parts.

## 4. Conclusions

This study investigated the MIG additive manufacturing method for fabricating 50-layer parts, and analyzed the effect of different duty ratios of double-pulse and single-pulse on the morphology, microstructure, and mechanical properties. The analysis results are as follows:

(1) Both single- and double-pulse modes could realize manufacturing thin walls, with the increase of the duty ratio from 35% to 65%, the heat input increased, and then the cladding amount of the molten pool increased, which resulted in a specimen height decrease from 61.92 to 58.62 mm and a width increase from 7.63 to 9.95 mm. However, the stirring effect of the double-pulse could increase the molten pool width and decrease the layer height.

(2) A good metal fusion was presented between the layers. The homogeneous structure including cellular and reticular austenite grains could be obtained by single-pulse and double-pulse strengthening processes. With the increase of the duty ratio, the microstructure became coarser. As the low frequency of the double-pulse had the effect of stirring the molten pool, the double-pulse part presented a finer microstructure.

(3) The Rockwell hardness of the bottom layers was larger than the top layers, and the hardness of both sides was smaller than the middle parts in the four samples. Increasing the duty ratio decreased the hardness little; however, the low frequency of the double-pulse could improve the hardness compared with the single-pulse one.

(4) The ultimate tensile strength of the single-pulse specimen was smaller than the double-pulse specimens. Increasing the duty ratio of the double-pulse from 35% to 65%, the average ultimate tensile strength decreased from 705 to 672 MPa.

Therefore, this study will provide an alternative additive manufacturing process for grain refinement and performance improvement and lay a foundation for further research and application of double-pulse welding arc additive manufacturing.

**Author Contributions:** Methodology, P.Y.; writing—original draft preparation, H.L. and W.W.; project administration and funding acquisition, P.Y.; data curation, H.L. and H.T.; writing—review and editing, providing ideas, P.Y. and W.W. All authors have read and agreed to the published version of the manuscript.

**Funding:** This research was funded by the National Natural Science Foundation Project of China, grant number 51805099; Project of Educational Commission of Guangdong Province of China, grant number 2020ZDZX2019; Guangzhou Science and Technology Plan Project, grant number 201805010001.

**Institutional Review Board Statement:** Not applicable.

**Informed Consent Statement:** Not applicable.

**Data Availability Statement:** Not applicable.

**Conflicts of Interest:** The authors declare no conflict of interest.

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
