# Peer review of "Influence of Duty Ratio and Current Mode on Robot 316L Stainless Steel Arc Additive Manufacturing"

_metals, doi:10.3390/met11030508_

Round 1

Reviewer 1 Report

Dear Authors,

Please, find my comments below.

1) I recommend extending the literature - quoting the last sentence of the first paragraph "... the research focus of many scholars and institutions in China and abroad".

2) I assume that the chemical composition (table 1) comes from the standard - from which?

3) Please unify the nomenclature: welding gun (page3, line 3) vs. welding torch (fig. 2), base plate (page 3, line 7,8,9) vs. substrate (fig. 2).

4) Necessary details of the devices (manufacturer, city, country).

5) Figure 7 shows a greater amount of spatter - pay attention to it in the text.

6) Adjust reference 2 to the requirements of the journal.

7) There are doubts in the reference to item 18 in the Materials and Methods chapter, which is a master's thesis whose title may indicate that Xiong, J. is the author of the research, and he is not included in the list of authors.

Author Response

Dear Reviewer:

Thank you very much for your comments/suggestions concerning our manuscript entitled “Influence of Duty Ratio and Current Mode on Robot Arc Additive Manufacturing” (Manuscript ID: metals-1123768).

The comments and suggestions are very valuable and helpful for us to improve the quality of the paper. We have carefully considered the comments/suggestions and have made a substantial revision to the manuscript. Revised places are marked by the "Track Changes" function in Microsoft Word in the text of the manuscript. Meanwhile, we have also prepared a list of changes and responses to the review comments as follows:

The results in this paper are interesting.  However, some parts need to be rewritten to make them clearer to the reader.

1) I recommend extending the literature - quoting the last sentence of the first paragraph "... the research focus of many scholars and institutions in China and abroad".

Response 1: Thank you for your kind comments and useful suggestions! We have added references followed the last sentence of the first paragraph.

2) I assume that the chemical composition (table 1) comes from the standard - from which?

Response 2:

It comes from the testing report of the manufacturer.

3) Please unify the nomenclature: welding gun (page3, line 3) vs. welding torch (fig. 2), base plate (page 3, line 7,8,9) vs. substrate (fig. 2).

Response 3: Thank you for your careful reviewing!

We have unified the nomenclature of “welding gun” and “base plate”.

4) Necessary details of the devices (manufacturer, city, country).

Response 4: Thank you for your useful suggestions! We have added the necessary details of the devices.

5) Figure 7 shows a greater amount of spatter - pay attention to it in the text.

Response 5: Thank you for your useful suggestions! We have added the spatter phenomenon.

6) Adjust reference 2 to the requirements of the journal.

Response 6: Thank you for your careful reviewing! We have revised the format of reference 2.

7) There are doubts in the reference to item 18 in the Materials and Methods chapter, which is a master's thesis whose title may indicate that Xiong, J. is the author of the research, and he is not included in the list of authors.

Response 7: Thank you very much for your kind comments and useful suggestions! We have added the author of Xiong in Materials and Methods chapter.

Best wishes!

Ping Yao, Hongyan Lin, Wei Wu and Heqing Tang

Reviewer 2 Report

Please find the attached report.

Author Response

Dear Reviewer:

Thank you very much for your comments/suggestions concerning our manuscript entitled “Influence of Duty Ratio and Current Mode on Robot Arc Additive Manufacturing” (Manuscript ID: metals-1123768).

The comments and suggestions are very valuable and helpful for us to improve the quality of the paper. We have carefully considered the comments/suggestions and have made a substantial revision to the manuscript. Revised places are marked by the "Track Changes" function in Microsoft Word in the text of the manuscript. Meanwhile, we have also prepared a list of changes and responses to the review comments as follows:

The current manuscript presents a characterization of the double-pulse and single-pulse MIG welding modes on the morphology, microstructure, and performance of stainless steel parts fabricated using WAAM. The topic is attractive and interesting results are presented. However, there are some issues that should be considered in the revised manuscript as follows:

  • The title is general, it is recommended to add the wire material used to make the title more compatible with the manuscript contents.

Response 1: Thank you for your useful suggestion! We have added the wire material of 316L Stainless Steel in the title.

  • A general grammatical review should be conducted taking into consideration the selection of some technical verbs and words.

Response 2: Thank you for your useful suggestion! We have selected some technical verbs and words.

  • The introduction section presents a good critical review. However, this section should be ended with a paragraph that declares the problem statement, direct application of the technique used in the current study, and a brief description of the experimental work including the expected contribution.

Response 3: Thank you for your useful suggestion! We have revised the last paragraph of the introduction.

  • The expression of “Duty ratio” should be well defined through the introduction section.

Response 4: Thank you for your useful suggestion! We have defined the expression of “Duty ratio” in the introduction.

  • Regarding the chemical composition of materials that are presented in Table 1, please illustrate if it was measured or it is related to a specific reference.

Response 5:

It comes from the testing report for base metal and welding wire of manufacturer.

  • The standard used to select the geometry of the tensile sample should be stated.

Response 6:

We have added the statement “The tensile samples designed in the test were smaller than the minimum size of the international standard sample. Therefore, all tensile sample sizes were reduced in proportion to the international standard; the tensile sample size is shown in Fig. 4”.

  • The current results and analysis characterizes the effect of single and double pulse modes on the quality of linear passes, how can we characterize the dimensional accuracy of more complex geometry parts?

Response 7:

This research was aimed at manufacturing thin-walled parts, and for complex parts, different key cross-sections was need to be cut for dimension measurement.

  • What is the effect of thermal residual stresses on the quality of parts for each mode (single/double)?

Response 8:

With the same process parameters and cooling time, single- and double-pulse modes of the approximately same heat input made the residual stress approximately same.

  • There should be more discussion about the validation of results through a more deep analysis to justify the obtained trends. The relationship between the microstructure observations and the mechanical properties should reflect and validate the presented analysis.

Response 9: Thank you for your kind comments and useful suggestions! We have added the relationship between the microstructure observations and the mechanical properties.

  • The conclusion section should clearly reflect the novelty and main contribution of the current study in addition to a summary of the main results.

Response 10:

Thank you for your kind comments and useful suggestions! We have revised the conclusions.

Best wishes!

Ping Yao, Hongyan Lin, Wei Wu and Heqing Tang

Reviewer 3 Report

The manuscript entitled: 'Influence of Duty Ratio and Current Mode on Robot Arc Additive Manufacturing' tries to study the importance of duty ratio and current moe on the RAAM process. I have the following concerns with the manuscript:

  • There is no clear scientific discussion connecting the influence of duty ratio and current mode with microstructure and mechanical properties.
  • For distinguishing coarser and finer microstructure, EBSD may be introduced to show the differences in the grain sizes.
  • XRD patterns may be introduced just to show that there is no phase transformation or phase changes observed as a function of duty ratio and current mode.
  • Space should be introduced between numbers and units (applicable also for scale bars).
  • Typos in the manuscript should be carefully rectified.

Author Response

Dear Reviewer:

Thank you very much for your comments/suggestions concerning our manuscript entitled “Influence of Duty Ratio and Current Mode on Robot Arc Additive Manufacturing” (Manuscript ID: metals-1123768).

The comments and suggestions are very valuable and helpful for us to improve the quality of the paper. We have carefully considered the comments/suggestions and have made a substantial revision to the manuscript. Revised places are marked by the "Track Changes" function in Microsoft Word in the text of the manuscript. Meanwhile, we have also prepared a list of changes and responses to the review comments as follows:

The manuscript entitled: 'Influence of Duty Ratio and Current Mode on Robot Arc Additive Manufacturing' tries to study the importance of duty ratio and current moe on the RAAM process. I have the following concerns with the manuscript:

  • There is no clear scientific discussion connecting the influence of duty ratio and current mode with microstructure and mechanical properties.

Response 1:

Thank you for your kind comments and useful suggestions! We have added the relationship between the microstructure observations and the mechanical properties.

  • For distinguishing coarser and finer microstructure, EBSD may be introduced to show the differences in the grain sizes. XRD patterns may be introduced just to show that there is no phase transformation or phase changes observed as a function of duty ratio and current mode.

Response 2:

Thank you for your useful suggestions of microstructural analysis methods!

  • Space should be introduced between numbers and units (applicable also for scale bars). Typos in the manuscript should be carefully rectified.

Response 3: Thank you for your kind comments and useful suggestions! We have added space between numbers and units.

Best wishes!

Ping Yao, Hongyan Lin, Wei Wu and Heqing Tang

Reviewer 4 Report

The current paper investigates fabrication of steel parts using single and double pulse MIG welding to study the effect of duty ratios and current modes on surface characteristics and performance. The authors conclude that the double pulse have stirring effects while the higher duty ratio decreased the height and increased the width.

The introduction is poorly written and is confusing, for example” The results demonstrate that double-pulse had stirring effect on the molten pool, and increasing the duty ratio from 35% to 65%, the specimen height decreased and the width increased, while the double-pulse mode had bigger width and smaller height than single-pulse mode.” This sentence have some issue and needs to be revised

Also what is meant by larger mechanical properties? This is very vague sentence in introduction!

“Moreover, the mechanical properties of the double-pulse specimens were larger than the single-pulse specimen, and increasing the duty ratio of the double-pulse specimens could decrease the hardness and the tensile strength.”

Saying could in introduction means the results are unclear and perhaps further testing is needed.

Table 1 needs referencing

Which standard was used for the specimen design?

Figure 5 does not provide any meaningful information at its current state.

There is very little discussion or explanation of the obtained results in the results and discussion section

The paper lacks novelty and is not suitable for reputable journal like metals

Author Response

Dear Reviewer:

Thank you very much for your comments/suggestions concerning our manuscript entitled “Influence of Duty Ratio and Current Mode on Robot Arc Additive Manufacturing” (Manuscript ID: metals-1123768).

The comments and suggestions are very valuable and helpful for us to improve the quality of the paper. We have carefully considered the comments/suggestions and have made a substantial revision to the manuscript. Revised places are marked by the "Track Changes" function in Microsoft Word in the text of the manuscript. Meanwhile, we have also prepared a list of changes and responses to the review comments as follows:

The manuscript entitled: 'Influence of Duty Ratio and Current Mode on Robot Arc Additive Manufacturing' tries to study the importance of duty ratio and current moe on the RAAM process. I have the following concerns with the manuscript:

The current paper investigates fabrication of steel parts using single and double pulse MIG welding to study the effect of duty ratios and current modes on surface characteristics and performance. The authors conclude that the double pulse have stirring effects while the higher duty ratio decreased the height and increased the width.

1) The introduction is poorly written and is confusing, for example” The results demonstrate that double-pulse had stirring effect on the molten pool, and increasing the duty ratio from 35% to 65%, the specimen height decreased and the width increased, while the double-pulse mode had bigger width and smaller height than single-pulse mode.” This sentence have some issue and needs to be revised.

Also what is meant by larger mechanical properties? This is very vague sentence in introduction!

“Moreover, the mechanical properties of the double-pulse specimens were larger than the single-pulse specimen, and increasing the duty ratio of the double-pulse specimens could decrease the hardness and the tensile strength.”

Saying could in introduction means the results are unclear and perhaps further testing is needed.

Response 1: Thank you for your kind comments and useful suggestions! We have revised the abstract.

2) Table 1 needs referencing

Which standard was used for the specimen design?

Response 2:

Table 1 comes from the testing report for base metal and welding wire of manufacturer.

3) Figure 5 does not provide any meaningful information at its current state.

Response 3:

Figure 5 shows the contour of single-pulse and double-pulse waveforms, as well as the contour of waveforms with different duty ratios. If the waveforms do not provide any meaningful information, we can delete them.

4) There is very little discussion or explanation of the obtained results in the results and discussion section

The paper lacks novelty and is not suitable for reputable journal like metals

Response 4: Thank you for your kind comments and useful suggestions! We have revised the conclusions.

Reviewer 5 Report

Dear authors,
the presented document is very interesting, however, it requires some necessary corrections.

  1. The "abstract" section should contain a broader description of the research presented in the document. The test results can possibly be written, but very briefly and generally. The abstract is intended to encourage reading, it is not a summary.
  2. The introduction section is written correctly.
  3. For what purpose is the chemical composition of the substrate given? The substrate is not used in the research. The test range of the tensile specimen is at least 15 mm from the substrate.
  4. Figure 4 should be signed separately.
  5. The microhardness chart should show where the sample is tested. Please add a photo of the sample with test sites. This is very important as a good quality photo will show the effect of fusion line on microhardness.
  6. When presenting the results of tensile strength, please include the curve of the monotonic tension for each case (Graph of stress versus strain). The mechanical properties should show the yield strength (0.2), the maximum tensile strength, the strain at break and the Young's modulus.
  7. In conclusion, please additionally present the results of each method for obtaining surface quality, microhardness and mechanical properties

Editorial note. Please keep uniformity when signing the drawings. Sub-items a), b) ... should be as, for example, in figure 10. Please correct the caption in figures 7, 15.

Kind Regards

Author Response

Dear Reviewer:

Thank you very much for your comments/suggestions concerning our manuscript entitled “Influence of Duty Ratio and Current Mode on Robot Arc Additive Manufacturing” (Manuscript ID: metals-1123768).

The comments and suggestions are very valuable and helpful for us to improve the quality of the paper. We have carefully considered the comments/suggestions and have made a substantial revision to the manuscript. Revised places are marked by the "Track Changes" function in Microsoft Word in the text of the manuscript. Meanwhile, we have also prepared a list of changes and responses to the review comments as follows:

The presented document is very interesting, however, it requires some necessary corrections.

1) The "abstract" section should contain a broader description of the research presented in the document. The test results can possibly be written, but very briefly and generally. The abstract is intended to encourage reading, it is not a summary.

Response 1: Thank you for your kind comments and useful suggestions! We have revised the abstract and the results.

2) The introduction section is written correctly.

For what purpose is the chemical composition of the substrate given? The substrate is not used in the research. The test range of the tensile specimen is at least 15 mm from the substrate.

Response 2: Thank you for your kind comments and useful suggestions! We have deleted the chemical composition of the substrate.

3) Figure 4 should be signed separately.

Response 3: Thank you for your useful suggestions! We have stated Figure 4 separately.

4) The microhardness chart should show where the sample is tested. Please add a photo of the sample with test sites. This is very important as a good quality photo will show the effect of fusion line on microhardness.

Response 4: Thank you for your useful suggestions! We have added a picture of the sample with hardness test sites.

5) When presenting the results of tensile strength, please include the curve of the monotonic tension for each case (Graph of stress versus strain). The mechanical properties should show the yield strength (0.2), the maximum tensile strength, the strain at break and the Young's modulus.

Response 5: Thank you for your kind comments and useful suggestions! We have presented the curves of stress versus strain.

6) In conclusion, please additionally present the results of each method for obtaining surface quality, microhardness and mechanical properties Editorial note.

Response 6: Thank you for your kind comments and useful suggestions! We have revised the conclusions.

7) Please keep uniformity when signing the drawings. Sub-items a), b) ... should be as, for example, in figure 10. Please correct the caption in figures 7, 15.

Response 7: Thank you for your careful reviewing! We have corrected the captions in figures 7, figure 15 and figure 16.

Best wishes!

Ping Yao, Hongyan Lin, Wei Wu and Heqing Tang

Round 2

Reviewer 1 Report

I have no comments

Author Response

Dear Editor:

Thank you very much for your review and best wishes for you!

Ping Yao, Hongyan Lin, Wei Wu and Heqing Tang

Reviewer 2 Report

Please find the attached report.

Author Response

Dear Editor:

Thank you very much for your letter and comments/suggestions concerning our manuscript entitled “Influence of Duty Ratio and Current Mode on Robot Arc Additive Manufacturing” (Manuscript ID: metals-1123768).

The comments and suggestions are very valuable and helpful for us to improve the quality of the paper. We have carefully considered the comments/suggestions and have made a substantial revision to the manuscript. Revised places are marked by the "Track Changes" function in Microsoft Word in the text of the manuscript. Meanwhile, we have also prepared a list of changes and responses to the review comments as follows:

The revised manuscript is improved and the authors covered most of the review comments and recommendations. However, there are some minor points that still need to be considered as follows:

  • By the end of the introduction section, there should be a fair declaration of the problem statement and the direct application of the current study.(2次)

Response 1: Thank you for your useful suggestion! We have revised the end of the introduction section.

  • In the materials and methods section; the following questions should be answered in the revised manuscript:
  • What kind of test and the equipment that was used for investigating the chemical composition of the wire material? 

Response: A NOVA NANO (FEI, Eindhoven, Netherlands) scanning electron microscope (SEM) 430 and map scanning method were used to investigate the chemical composition of the wire material.

  • What is the targeted layer thickness in the current work? 

Response: The targeted layer thickness is 1.25 mm in the current work.

  • What is the reference from where the process parameters were selected?

Response: The process parameters were selected by reference [18] and single pass welding process experiment.

  • Figure 5 does not add a significant data; it is recommended to be deleted.

Response 3: Thank you for your useful suggestion! We have deleted Figure 5.

  • What is the standard deviation of the data presented in Figures 9 and 10?

Response 4: Figures 9 and 10 present the height and width of each point, therefore, there is no standard deviation in the bar chart. The standard deviation of the mean value was given in the paper that “It can be seen from Fig. 6 and Fig. 7 that the average height of A, B, C and D part was 61.92 ± 0.93, 61.07 ± 0.48, 58.62 ± 0.20 and 63.52 ± 0.76 mm, respectively, and the average width was 7.63 ± 1.41, 8.73 ± 1.38, 9.95 ± 1.08 and 7.67 ± 0.67 mm, respectively”.

  • The results and discussion section still lacks the technical references that should directly support the analytical data.

Response 5: Thank you for your useful suggestion! We have added some technical references in results and discussion section.

  • The conclusion section still needs to clearly reflect the novelty and direct application of the current work.

Response 6: Thank you for your useful suggestion! We have revised the conclusion section.

Best wishes!

Ping Yao, Hongyan Lin, Wei Wu and Heqing Tang

Reviewer 3 Report

The authors have not addressed the comments in a satisfactory manner and all my previous comments still hold valid:

(1) Stronger scientific discussion needed.

(2) XRD and EBSD should be introduced.

(3) Typos need to be rectified. For Instance. Fig. 15 - scale bars space should be introduced between number and units.

Author Response

Dear Editor:

Thank you very much for your letter and comments/suggestions concerning our manuscript entitled “Influence of Duty Ratio and Current Mode on Robot Arc Additive Manufacturing” (Manuscript ID: metals-1123768).

The comments and suggestions are very valuable and helpful for us to improve the quality of the paper. We have carefully considered the comments/suggestions and have made a substantial revision to the manuscript. Revised places are marked by the "Track Changes" function in Microsoft Word in the text of the manuscript. Meanwhile, we have also prepared a list of changes and responses to the review comments as follows:

The authors have not addressed the comments in a satisfactory manner and all my previous comments still hold valid:

  • Stronger scientific discussion needed.

Response 1:

Thank you for your kind comments and useful suggestions! We have revised the discussion.

  • XRD and EBSD should be introduced.

Response 2:

Thank you for your useful suggestions of microstructural analysis methods! We have calculated the secondary dendrite spacing to present the grain size and showed them in the paper.

  • Typos need to be rectified. For Instance. Fig. 15 - scale bars space should be introduced between number and units.

Response 3: Thank you for your kind comments and useful suggestions!

We had added the space of scale bars in Figure 14-18.

Best wishes!

Ping Yao, Hongyan Lin, Wei Wu and Heqing Tang

Reviewer 4 Report

“Furthermore, the tensile samples designed in the test were smaller than the minimum size of the international standard sample. Therefore, all tensile sample sizes were reduced in proportion to the international standard; the tensile sample size and is shown in Fig..” did this have any effect on the results?

Again the authors are still using two figures beside each other instead of using a and b.. this must be changed before the paper can be considered in the journal

Please consider reviewing the results and discussion section. The results are merely described and is limited to comparing the experimental observation. The authors are encouraged to include a discussion section and critically discuss the observations from this investigation with existing literature.

I read the manuscript again but i didnt see any critical discussion whatsoever, please explain the phenomena and mechanisms you describe and why they happened. 

For example see this :  Furthermore, the hardness of the bottom layers was larger than the top layers, and the hardness of both sides was smaller than the middle parts, this was for the higher cooling rate of the bottom part from the base plate and the sides from the atmosphere.

you mentioned why it was large but then you need to followup of why it is larger and if this was also observed in previous studies or not, then explain how the higher cooling rate affect the hardness of this material specifically..

Author Response

Dear  Reviewer:

Thank you very much for your letter and comments/suggestions concerning our manuscript entitled “Influence of Duty Ratio and Current Mode on Robot Arc Additive Manufacturing” (Manuscript ID: metals-1123768).

The comments and suggestions are very valuable and helpful for us to improve the quality of the paper. We have carefully considered the comments/suggestions and have made a substantial revision to the manuscript. Revised places are marked by the "Track Changes" function in Microsoft Word in the text of the manuscript. Meanwhile, we have also prepared a list of changes and responses to the review comments as follows:

1) “Furthermore, the tensile samples designed in the test were smaller than the minimum size of the international standard sample. Therefore, all tensile sample sizes were reduced in proportion to the international standard; the tensile sample size and is shown in Fig..” did this have any effect on the results?

Response 1: Although the specimens are non-standard, the comparison between four samples had no effect on the results of this study.

2) Again the authors are still using two figures beside each other instead of using a and b.. this must be changed before the paper can be considered in the journal

Response 2: Thank you for your useful suggestions, we have combined figures 3-5 into figure 3, and figures 11-14 into figure 9.

3) Please consider reviewing the results and discussion section. The results are merely described and is limited to comparing the experimental observation. The authors are encouraged to include a discussion section and critically discuss the observations from this investigation with existing literature. I read the manuscript again but i didnt see any critical discussion whatsoever, please explain the phenomena and mechanisms you describe and why they happened. For example see this :  Furthermore, the hardness of the bottom layers was larger than the top layers, and the hardness of both sides was smaller than the middle parts, this was for the higher cooling rate of the bottom part from the base plate and the sides from the atmosphere. you mentioned why it was large but then you need to followup of why it is larger and if this was also observed in previous studies or not, then explain how the higher cooling rate affect the hardness of this material specifically.

Response 3: Thank you for your kind comments and useful suggestions! We have revised the discussion.

Best wishes!

Ping Yao, Hongyan Lin, Wei Wu and Heqing Tang

Reviewer 5 Report

Dear authors, I accept the amendments made.
In my opinion, the article is suitable for publication.
I wish you good luck and a lot of scientific success

Author Response

Dear  Reviewer:

Thank you very much for your review and encouragement, best wishes for you!

Ping Yao, Hongyan Lin, Wei Wu and Heqing Tang

Round 3

Reviewer 3 Report

I am disappointed with the author's response. 

  • A strong scientific discussion - not addressed
  • XRD and EBSD - not included
  • Only typos were rectified and still, the manuscript has typos.

I cannot accept the manuscript to be published in this condition.

Author Response

Dear Reviewer:

Thank you very much for your letter and comments/suggestions concerning our manuscript entitled “Influence of Duty Ratio and Current Mode on Robot Arc Additive Manufacturing” (Manuscript ID: metals-1123768).

The comments and suggestions are very valuable and helpful for us to improve the quality of the paper. We have carefully considered the comments/suggestions and have made a substantial revision to the manuscript. Revised places are marked by the "Track Changes" function in Microsoft Word in the text of the manuscript. Meanwhile, we have also prepared a list of changes and responses to the review comments as follows:

  • A strong scientific discussion - not addressed.

Response 1:

Thank you for your kind comments and useful suggestions! We have made a revision in the discussion part.

  • XRD and EBSD not included

Response 2:

Thank you for your command. We are sorry that we do not have the conditions to do these tests at present; therefore, the grain size is reflected by the measured value of the secondary dendrite spacing of columnar grains.

  • Only typos were rectified and still, the manuscript has typos.

Response 3: Thank you for your kind comments!

We had careful check the typos and revised, such as “I” in Table 2, and caption of Figure 3, Figure 4 and Figure 11.

Finally, we appreciate very much for your time in reviewing our manuscript.

Best wishes!

Ping Yao, Hongyan Lin, Wei Wu and Heqing Tang

Reviewer 4 Report

Authors answered all questions

Author Response

Dear Reviewer:

Thank you very much for your letter and comments/suggestions concerning our manuscript entitled “Influence of Duty Ratio and Current Mode on Robot Arc Additive Manufacturing” (Manuscript ID: metals-1123768).

The comments and suggestions are very valuable and helpful for us to improve the quality of the paper.

  • Authors answered all questions

Response: Thank you very much for your review and best wishes for you!

Ping Yao, Hongyan Lin, Wei Wu and Heqing Tang

Round 4

Reviewer 3 Report

The authors still have not addressed all my comments.